# Fluorescence Detecting of Paraquat and Diquat Using Host–Guest Chemistry with a Fluorophore-Pendant Calix[6]arene

**DOI:** 10.3390/s23031120

**Published:** 2023-01-18

**Authors:** Ermanno Vasca, Francesco Siano, Tonino Caruso

**Affiliations:** 1Department of Chemistry and Biology, University of Salerno, 84084 Fisciano, Italy; 2Institute of Food Science, National Research Council, 83100 Avellino, Italy

**Keywords:** fluorescence quenching, herbicides, paraquat, diquat, molecular recognition

## Abstract

Paraquat (PQ) and diquat (DQ), some of the most widely used herbicides in the world, both present a high mortality index after intentional exposure. In this paper, a fluorescence sensing method for PQ and DQ, based on host–guest molecular recognition, is proposed. Calix[6]arene derivatives containing anthracene or naphthalene as pendant fluorophore at their lower rim recognize DQ and PQ in hydroalcoholic solution with a broad linear response range at the μg L^−1^ level concentration. The linear response ranges were found from 1.0 to 18 μg L^−1^ with the detection limit of 31 ng L^−1^ for paraquat, and from 1.0 to 44 μg L^−1^ with the detection limit of 0.16 μg L^−1^ for diquat. The recognition process is detected by following the decrease in the fluorescence emission consequent to complexation. The proposed quenching method has been applied to the determination of paraquat in drinking water samples.

## 1. Introduction

Paraquat (1,1′-dimethyl-4,4′-bipyridylium dichloride, PQ) and diquat (1,1′-ethylene-2,2′-bipyridilium dibromide, DQ) (Figure 1) are non-selective and nonsystematic contact herbicides widely used in agriculture to control broadleaf and grassy weeds. The use of these herbicides is important because weeds compete vigorously with crops for water, light and other nutrients. However, both contain a diquaternary bipyridyl unit that is responsible for their herbicidal and toxicological properties, while the halogen anions have few toxic effects [1,2,3]. Because of their highly toxic effects on humans, their use is “restricted” in the USA and banned in the EU and elsewhere [4,5,6].

The detection and quantitative determination of bipyridilium herbicides are somewhat difficult mainly because they are cationic molecules. As in the case of quaternary amines, for chromatography and hyphenated techniques [7,8,9,10,11], their inherent high polarity and positive charge make it necessary to use ion pairing additives if reversed-phase columns are used. Numerous other methods have been developed to monitor and control these herbicides in the environment, water, food and clinical samples, including capillary electrophoresis [12], voltammetry [13,14] and immunoassay [15,16]. Fluorescence spectroscopy is known for its sensitivity, high specificity, selectivity, speed, simplicity and low cost as compared to other analytical techniques [17]. Due to the high levels of sensitivity and particularly their ability to be used for temporal and spatial sampling for in vivo imaging applications, fluorescent chemosensors based on molecular recognition have been widely applied in a variety of fields such as biology, physiology, pharmacology and environmental sciences [18,19]. Variations in the fluorescence spectrum of a guest molecule (G), consequent to complexation, are usually used to detect trace amounts of a host analyte (H).

However, few fluorescence methods have been developed for paraquat and diquat detection [20,21,22,23,24]. Recently, a novel non-covalently linked photoreactive dyad was synthetized by applying supramolecular assembly based on a calixarene host–guest chemistry [25], in which the fluorescent probes, naphthalene or anthracene, are covalently attached to the host (Figure 2).

Naphthalene and anthracene are indeed the most widely used fluorogenic units in the synthesis of fluorescent calixarene host [26,27,28]. It was demonstrated that simple ca-lixarenes can be efficient wheels toward dialkylammonium axles by exploiting the inducing effect of the weakly coordinating tetrakis[3,5-bis-(trifluoromethyl)phenyl]borate (TFPB-) [29]. Threading of calix[6]arene macrocycles to the TFPB salt of dialkylammonium cations can be a valid tool to prepare a fluorescent pseudo-rotaxane, a host–guest system composed minimally of a threadlike molecule “as axles” surrounded by a macrocycle [30].

Based on this strategy, diquat dibromide (DQ) and paraquat dichloride (PQ) are expected to behave similarly to the dialkylammonium axles. In the present study, the chemosensory behavior of these pseudo-rotaxanes composed by a new calixarene hosts with paraquat and diquat was investigated by fluorescence spectroscopy, in order to develop a sensitive fluorescent detection method for PQ and DQ in water samples.

## 2. Materials and Methods

All solvents used in this study for fluorescence measurements were of spectrophotometric grade. Chloroform, methanol and deionized water were purchased from Sigma-Aldrich Corporation. All reagents used in this study were of analytical grade. Diquat dibromide monohydrate (DQ), paraquat dichloride hydrate (PQ) and the interfering species investigated in this work, carbaryl, atrazine and triclopyr, all were purchased from Sigma-Aldrich. Anion and cations were prepared from ionic salts (NaCl, KNO_3_, Na_2_CO_3_, Na_2_SO_4_, Na_3_PO_4_, MgCl_2_, CaCO_3_ and ZnSO_4_·7H_2_O) purchased from Merck. The calixarene hosts 1a and 1b (Figure 2) used in this study were synthesized and purified by HPLC by a published method [25,31].

To investigate the complexing ability of 1a and 1b toward DQ and PQ dications, fluorometric titration experiments were performed. The association constants of 1a and 1b with PQ and DQ cannot be determined by UV–Vis spectroscopy, because spectral changes upon host–guest interaction are too small. The chemosensor behavior of the new host with PQ and DQ was investigated by fluorescence spectroscopy in a chloroform–methanol 1:1 solvent mixture. UV–Vis spectra were recorded on a Varian Cary 50 UV–Vis Spectrophotometer, in the 200–800 nm wavelength interval, to determine the maximum of the absorption spectrum of the fluorophores 1a and 1b. This allowed us to select the excitation frequency for the successive fluorescence experiments, according to Kasha’s rule [32]. The most intense spectra were obtained with an excitation energy slightly higher than the one corresponding to the absorption maximum (ca 50 eV). Fluorescence measurements were performed on a Varian Cary Eclipse Fluorescence Spectrophotometer equipped with a Varian Cary Single Cell Peltier to conduct measurements at a controlled temperature. A Thermo Scientific UltiMate 3000 Binary semi-preparative system, equipped with an Agilent RP Prep-C18 Scalar column, 100 Å, 4.6 mm × 150 mm, 10 µm and a Diode array detector, was used for purifying the host compounds [31]. A Metrohm 787 KF Titrino Karl Fisher Titration System was used to determine the water content in the solid samples [33]. A Metrohm 715 Dosimat was used to dose a titration solution with the precision of ±0.001 mL. A Mettler Toledo XS105DR analytical balance was used to weigh with a precision of ±0.01 mg. To value if the herbicide concentration depends on the matrix of the sample, a Perkin-Elmer ICP-OES Optima 3000 Dual View Spectrophotometer was used to determine the metal concentration in a tap water sample, and Thermo-Dionex Aquion Ion Chromatography was used to determine the anion concentration in a tested tap water sample.

## 3. Results and Discussion

### 3.1. Determination of Association Constants by Fluorometric Titration

The most common approach for quantifying interactions in supramolecular chemistry is a guest–host titration, noting the changes in some physical property through NMR, UV–Vis, fluorescence or other techniques. For this purpose, to avoid the volume variation, H-G solutions were prepared by adding equal volumes of the solutions with the same host concentration to different volumes of the paraquat or diquat standard solution. The final volume of each solution was made up to the same total volume by fixing with the solvent mixture. All the solutions were prepared in chloroform–methanol 1:1 mixture solution, equilibrated in a sealed vial at 25 °C for 24 h before measurements. The experimental values are reported in Table 1, Table 2, Table 3 and Table 4.

The emission spectra were recorded using excitation wavelengths of 285 nm for 1a and 263 nm for 1b, characteristic wavelengths of maximum absorption for the appended naphthalene or anthracene chromophore [34,35]. The addition of a highly concentrated chloroform–methanol solution of paraquat or diquat to a chloroform–methanol solution of the 1a or 1b host resulted in a drastic decrease in the intensity of the host emission peak (Figure 3 and Figure 4). Similar results were observed in the diquat titration (Figure 5 and Figure 6).

Based on the fluorescence measurements, the extended Benesi–Hildebrand equation (Equation (1)) can be used to calculate the association constants (*K*) for the 1:1 model, considering that under the experimental conditions employed, the final concentration of the guest herbicide is much larger than that of the calixarene host, i.e., [G] >> [H] [36,37]. In this equation, Δ*F* denotes the changes in the fluorescence intensity as *F*_0_ − *F*, in which *F*_0_ is the value of the maximum fluorescence spectrum of the host, and Δ*ε* denotes the molar extinction coefficient of the H-G complex.
(1)$$\frac{\left[H\right]\left[G\right]}{\left[\Delta F\right]}=\frac{1}{K\Delta \varepsilon }+\frac{\left[G\right]}{\Delta \varepsilon }$$,

Figure 7 illustrates the results of such a treatment for the H-G interaction, where the calculated values of [H][G]/[Δ*F*] are reported against the molar concentrations of G, affording an adequate linear relationship.

The values of the constants for the 1:1 complex calculated from the slope and intercept of [H][G]/Δ*F* versus [G] plots are reported in Table 5.

The binding constant 1:1 of PQ–host is higher than that of DQ–host probably due to the planar structure of PQ compared to DQ. Calculations are being made to determine the interaction energy between the aromatic units of calix and herbicide guests, as in the case of stacking interactions between two nucleobases [38]. Moreover, the fit of the experimental data with non-linear least squares was found to be independent of the *K*_12_ value. Therefore, the 1:1 model prevails [39].

### 3.2. Stern–Volmer Plot

Fluorescence quenching refers to any process that decreases the fluorescence intensity of the sample. A variety of molecular interactions can result in quenching, such as excited-state reactions, molecular rearrangements, energy transfer, ground-state complex formation and collisional quenching. The quenching resulting from collisional encounters between the fluorophore and quencher is called collisional or dynamic quenching. The quenching resulting from a non-fluorescent complex formation between the fluorophore and the quencher is called static quenching. Both static and dynamic quenching require molecular contact between the fluorophore and quencher. Collisional quenching of fluorescence is described by the Stern–Volmer Equation (2):(2)$$\frac{F_{0}}{F}=1+K_{D}\left[Q\right]$$,

In Equation (2), *F*_0_ and *F* are the fluorescence intensities in the absence and presence of quencher, respectively, and [Q] is the concentration of quencher, i.e., paraquat or diquat in this work. *K_D_* is the Stern–Volmer quenching constant that depends on a bimolecular quenching constant and the lifetime of the fluorophore in the absence of quencher. It is important to recognize that the observation of a linear Stern–Volmer plot does not prove that collisional quenching of fluorescence has occurred.

The static quenching also results in linear Stern–Volmer plots, as described by the Stern–Volmer Equation (3):(3)$$\frac{F_{0}}{F}=1+K_{S}\left[Q\right]$$,

The linear dependence of *F*_0_/*F* on [Q] is identical to that observed for dynamic quenching, except that the static quenching constant *K_S_* now coincides with the association constant H-G [35].

The Stern–Volmer plot of *F*_0_/*F* versus [Q] is shown in Figure 8, for paraquat or diquat as the quencher. The ratios *F*_0_/*F* display a pronounced upward curvature at high concentrations of the quencher. This characteristic feature of the Stern–Volmer plot is typical for quenching that results both by collisions and by complex formation with the same quencher [35,40,41,42]. A modified form of the Stern–Volmer Equation (4) accounts for the upward curvature observed when both static and dynamic quenching occur for the same fluorophore [35] (pp. 282–284):(4)$$\frac{F_{0}}{F}=\left(1+K_{D}\left[Q\right]\right)\cdot \left(1+K_{S}\left[Q\right]\right)$$,

Moreover, we also observed strict linearity of the *F*_0_/*F* Stern–Volmer plot until 60 and 90 μg L^−1^ for paraquat and diquat, respectively, as reported in Figure 9. As discussed above, when the Stern–Volmer plot is linear, only one type of quenching occurs, probably a static quenching. Indeed, by using Equation (3), the values for the H-G binding constants can be calculated [35]. These values (see Table 6) are similar to the those determined by fitting the spectral data according to the Benesi–Hildebrand equation (Figure 7, Table 5) [39]. Therefore, dynamic quenching is negligible at a low guest concentration [35] (p. 282), when the formation of the strong 1:1 complex prevails.

### 3.3. Fluorescent Detection of Paraquat and Diquat

The resulting emission spectra in the presence of paraquat or diquat show that the fluorescence is already quenched efficiently at μg L^−1^ concentrations of guest, and show a linear correlation between fluorescence change Δ*F* = *F*_0_ − *F* and the added guest concentration. *F*_0_ indicates the fluorescence intensity at [PQ] = 0 μg L^−1^ or [DQ] = 0 μg L^−1^ at 336 nm for naphthalene fluorophore 1a and at 416 nm for anthracene fluorophore 1b. In Figure 10, this adequate linearity at μg L^−1^ concentration is evident, indicating that the probes 1a and 1b can quantitatively detect these herbicides at ppb concentrations.

To verify if it is possible to determine very low concentrations of these herbicides by fluorescent recognition, the detection limit was calculated from the fluorescence titration data, as defined by IUPAC [43,44]. The fluorescence spectra of fluorophores were measured ten times and the standard deviation of blank measurement was achieved at the prefixed wavelength. To gain the slope of the curve *F* against concentration, the fluorescence intensity data collected at the maximum were plotted against the concentration of paraquat or diquat. So, the detection limit was calculated with the following Equation (5):Detection limit = 3 *σ*/*m*,(5)
where *σ* is the standard deviation of blank measurement, and *m* is the slope of the linear equation that relates the maximum fluorescence intensity to the guest concentrations.

The calculated detection limits reported in Table 7 are comparable or better than those obtained with other sensors for determination of these analytes in water without sample preconcentration [23,24]. The probes 1b-PQ and 1b-DQ present the best prospects for herbicide determination, especially the 1b-PQ probe, which was found to be the most sensitive under the experimental conditions. Moreover, the calix-appended anthracene allows us to determine an herbicide concentration better than the calix-appended naphthalene fluorophore.

For drinking water, the U.S. Environmental Protection Agency (EPA) has established a maximum contaminant level of 20 μg L^−1^ for diquat and a desired goal of 3 μg L^−1^ for paraquat (not EPA-regulated). The European Union has not regulated the levels of these compounds specifically in drinking water and continues to apply the value of 0.1 μg L^−1^ expected for all pesticides [45,46]. Therefore, the proposed procedure method could be applied for the determination of these herbicides in water samples, by using hydroalcoholic solutions to allow host–guest solubility. However, it is also expected that the presence of the more polar solvent can play a significantly larger role in the electrostatic interactions and decrease or increase the fluorescence signal [47,48].

### 3.4. Validation of the Fluorescent Detection in Water Environment

A sample of tap water was spiked with different amounts of paraquat or diquat, ranging from 3 to 75 μg L^−1^, by using a standard solution of guest. All the samples were also treated with EDTA, 1 mg/mL, to suppress the possible interference of metal ions [49]. To avoid solvatochromic effects on fluorescence emission spectra [50], the method of standard additions was used [51], by adding methanol solutions of paraquat (or diquat) and 1b to several aliquots of the same volume of tap water, spiked at 1.0 μg L^−1^ of total solution. The resulting hydroalcoholic solutions must have a concentration of 60% *v*/*v* in a compromise between signal peak height and adopting the highest concentration of water possible. Therefore, a concentration of sample water of 40% was selected for the experiments. Table 8 reports the experimental data for the paraquat test, plotted in Figure 11 as Δ*F* = *F*_0_ − *F*, in which the *F*_0_ value was measured in a non-spiked sample obtained from ultrapure, HPLC-grade water. The straight line intercepts the concentration axis at the expected value, in agreement with the extrapolated value of (1.1 ± 0.2) μg L^−1^ obtained from the standard addition method. It is evident that all species at a concentration normally found in tap water did not interfere (see Section 3.5). Similarly, a diquat test confirmed the spike concentration in the same sample of water used for the paraquat test. The experimental data for DQ test are gathered in Table 9 and Figure 12.

### 3.5. Tolerance Limit for Possible Interfering Species

As discussed, it seems evident that all species normally found in tap water did not interfere in the paraquat or diquat determination using a standard addition method. The detected species in our sample and their concentrations are reported in Table 10.

In this section, the effect of these water saline components was studied, but at a higher concentration to determine the tolerance concentration, defined as the concentration that did not vary by more than 5% for the analytical signal. Combining the salts NaCl, KNO_3_, Na_2_CO_3_, Na_2_SO_4_, Na_3_PO_4_, MgCl_2_, CaCO_3_ and ZnSO_4_·7H_2_O, a sample of 10.0 mL of tap water was fortified 10× in these anions and cations and the difference in the maximum fluorescence intensity, *F*_0_, was determined with respect to the *F* value recorded before the addition of these salts. The signal variation was calculated according to Equation (6):(6)$$signal\;variation=\frac{\left|F_{0}-F\right|}{F_{0}},\%$$

The addition did not produce any signal change greater than 5% at a herbicide concentration of 1.0 μg L^−1^. Instead, ion concentrations higher than 10 times can produce a signal variation depending on the concentration and the kind of species. Actually, the mineral content of typical tap or drinking water is within the tested range [52,53].

Potential cross-interferences were also evaluated by using Equation (6), comparing *F*_0_, the maximum fluorescence intensity of a paraquat solution, with *F*, the maximum fluorescence intensity of an equimolar solution of paraquat and diquat. The last solution had the same concentration as the first solution, in the linear range from 1.0 to 10.0 μg L^−1^. At the same concentration of paraquat, no significant interferences were observed for DQ in this range (Table 11), probably because the interaction for the 1b-PQ probe is preferred (Table 5). This probe also shows better calibration sensitivity with respect to the 1b-DQ probe (see Figure 10). Only when the diquat concentration is increased by more than 18 times with respect to the paraquat level, the variation in the fluorescence signal will be more than 5%. The most serious interference will be the presence of paraquat in the diquat direct determination, because of the higher affinity of 1a-1b hosts for PQ guest (Table 5). In this case, the PQ interference can be higher than 5%.

The standard addition method revisited can lead to adequate precision provided that a large excess of the analyte G was used and [G] > [H] > [interferent guest] [54,55]. Indeed, a large excess of the added guest G with respect to the interferent guest shifts the equilibrium towards the formation of host–guest favored by entropy. The result will be a distribution of points only slightly curved at low concentrations of G, when [G] < [H]. Instead, the calibration line achieved in the range [G] ≥ [H] and extrapolated to zero signal provides a precise and accurate result. Experimental parameters, such as the increment size and the number of additions, must obviously be good enough to obtain a straight line (Table 12) [56].

By plotting the Δ*F* values reported in Table 12 against diquat concentration, [DQ], the intersection with the x-axis by extrapolation of the straight line obtained in the range [DQ] ≫ [1b] affords the expected diquat concentration of (6.1 ± 0.3) μg L^−1^, also in the presence of paraquat at the same concentration.

## 4. Conclusions

In this work, a new fluorescent sensing method for paraquat and diquat was developed, and we focused on the improved determination of paraquat in tap and drinking waters. A calix[6]arene derivative containing anthracene or naphthalene as a pendant fluorophore at a lower rim can detect these herbicides in a very low concentration through the host–guest interaction. The fluorescence intensity change showed an adequate linear relationship with the herbicide concentrations. The linear response ranges were found from 1.0 to 18 μg L^−1^ with the detection limit of 31 ng L^−1^, and from 1.0 to 44 μg L^−1^ with the detection limit of 0.16 μg L^−1^ for paraquat and diquat, respectively.

## Figures and Tables

**Figure 1 sensors-23-01120-f001:**
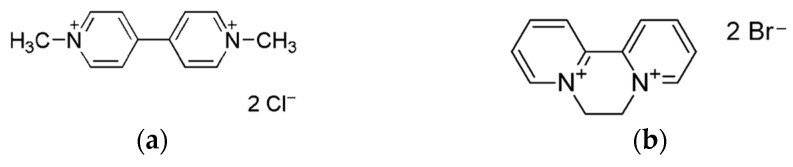
(**a**) Paraquat dichloride, PQ. (**b**) Diquat dibromide, DQ.

**Figure 2 sensors-23-01120-f002:**
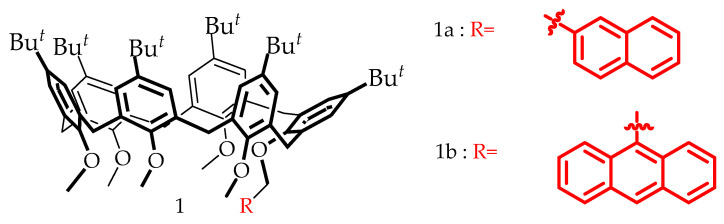
Calix[6]arene hosts 1a and 1b with fluorophores covalently linked.

**Figure 3 sensors-23-01120-f003:**
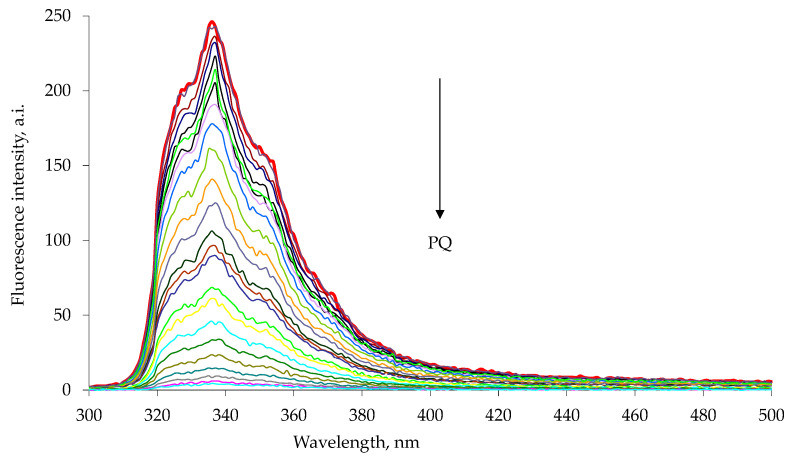
Fluorescence quenching of 1a host with increasing paraquat concentration, PQ, in chloroform–methanol 1:1 solution at 25 °C. The experimental data are reported in Table 1. The emission spectra were measured with excitation at 285 nm. Each spectrum is reported with different color line (Entry 1–25 Table 1).

**Figure 4 sensors-23-01120-f004:**
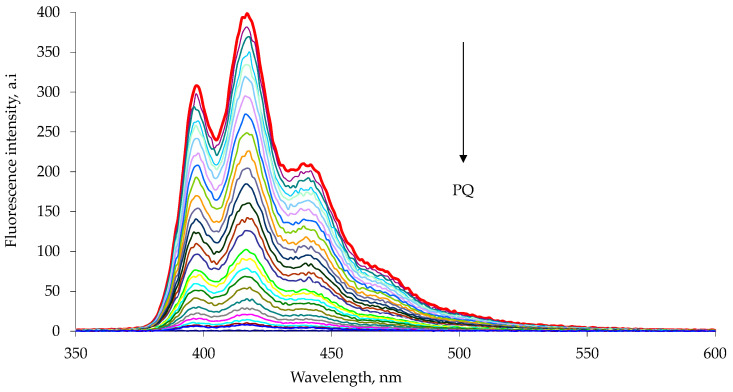
Fluorescence quenching of 1b host with increasing paraquat concentration, PQ, in chloroform–methanol 1:1 solution at 25 °C. The experimental data are reported in Table 2. The emission spectra were measured with excitation at 263 nm. Each spectrum is reported with different color line (Entry 1–25 Table 2).

**Figure 5 sensors-23-01120-f005:**
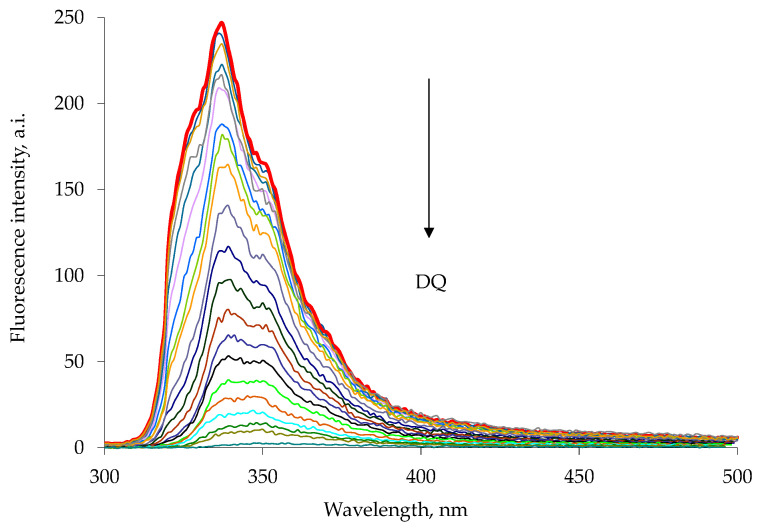
Fluorescence quenching of 1a host with increasing diquat concentration, DQ, in chloroform–methanol 1:1 solution at 25 °C. The emission spectra were measured with excitation at 285 nm. Each spectrum is reported with different color line (Entry 1–18 Table 3).

**Figure 6 sensors-23-01120-f006:**
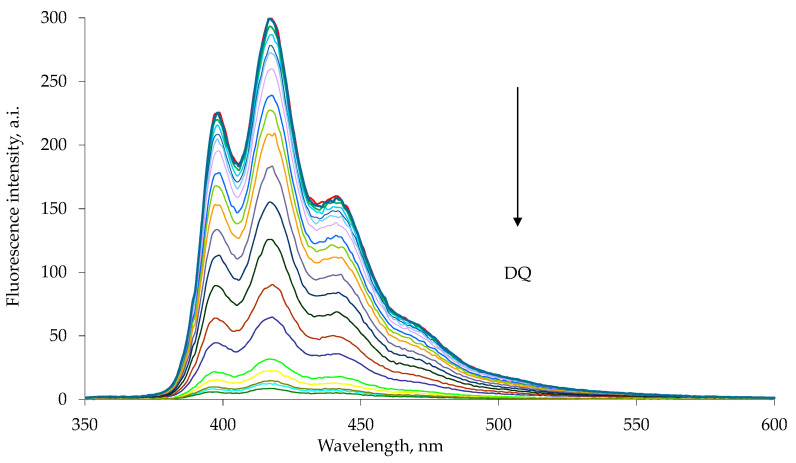
Fluorescence quenching of 1b host with increasing diquat concentration in chloroform–methanol 1:1 solution at 25 °C. The emission spectra were measured with excitation at 263 nm. Each spectrum is reported with different color line (Entry 1–20 Table 4).

**Figure 7 sensors-23-01120-f007:**
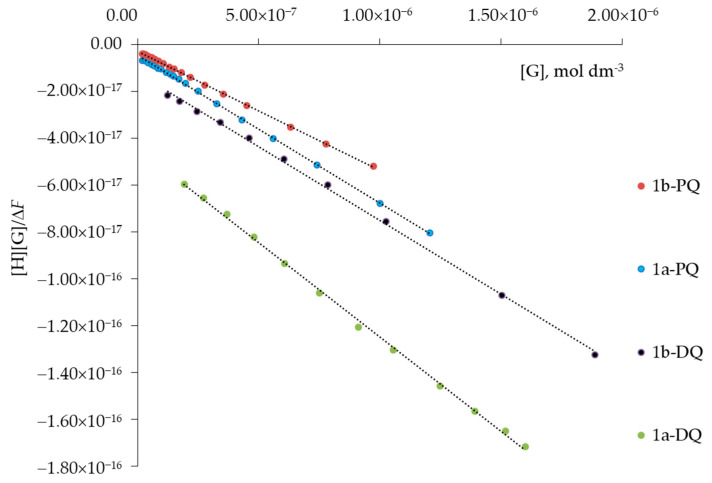
Typical plot of [H][G]/Δ*F* versus [G], mol dm^−3^ for the complexation of host 1a (or 1b) with the guest (G), paraquat (PQ) or diquat (DQ). The dashed lines represent the fitted function by using Equation (1).

**Figure 8 sensors-23-01120-f008:**
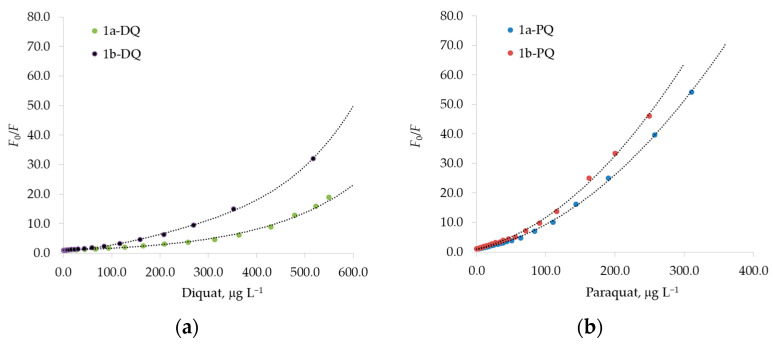
A modified Stern–Volmer plot of *F*_0_/*F* against increasing concentration of quencher guests. (**a**) The experimental behavior for the diquat, DQ, as quencher of 1a and 1b fluorophores. (**b**) The experimental behavior for the paraquat, PQ, as quencher of 1a and 1b fluorophores. The dashed lines represent the curvature provided by Equation (4). The excitation wavelength was set to 285 nm and 263 nm, and the fluorescence value *F* was measured at 336 nm and 416 nm for the fluorophores 1a and 1b, respectively. *F*_0_ is the fluorescence intensity in the absence of quencher PQ or DQ.

**Figure 9 sensors-23-01120-f009:**
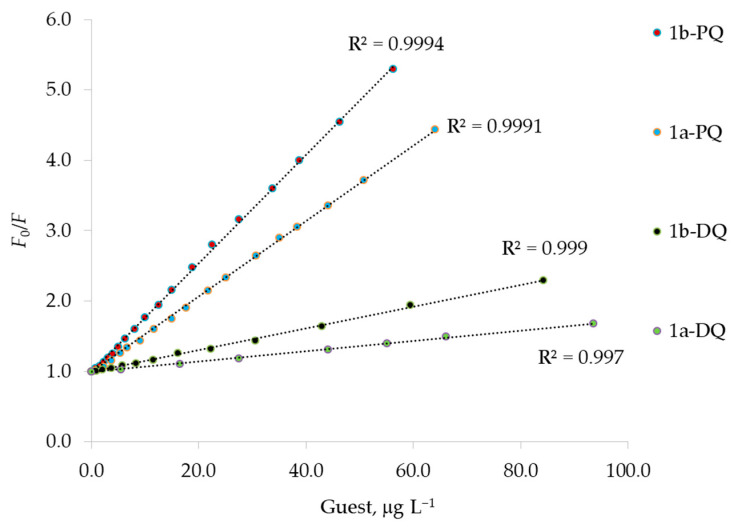
Stern–Volmer plots for the fluorescence quenching of 1a and 1b host with paraquat, PQ, and diquat, DQ, as guests. The excitation wavelength was set to 285 nm and 263 nm, and the fluorescence value *F* was measured at 336 nm and 416 nm for the fluorophores 1a and 1b, respectively. *F*_0_ is the fluorescence intensity in the absence of quencher PQ or DQ at 336 nm or 416 nm.

**Figure 10 sensors-23-01120-f010:**
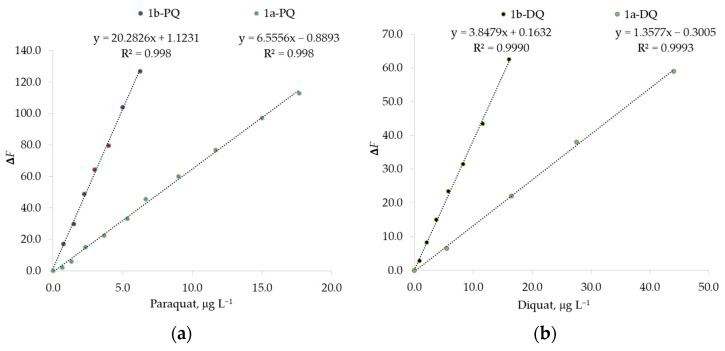
Fluorescence change Δ*F* = *F*_0_ − *F* of fluorophores 1a and 1b against (**a**) paraquat, PQ, or (**b**) diquat, DQ, concentrations in chloroform–methanol solution 1:1. *F*_0_ indicates the fluorescence at [PQ] = 0 μg L^−1^ or [DQ] = 0 μg L^−1^ at 336 nm for naphthalene fluorophore 1a and at 416 nm for anthracene fluorophore 1b.

**Figure 11 sensors-23-01120-f011:**
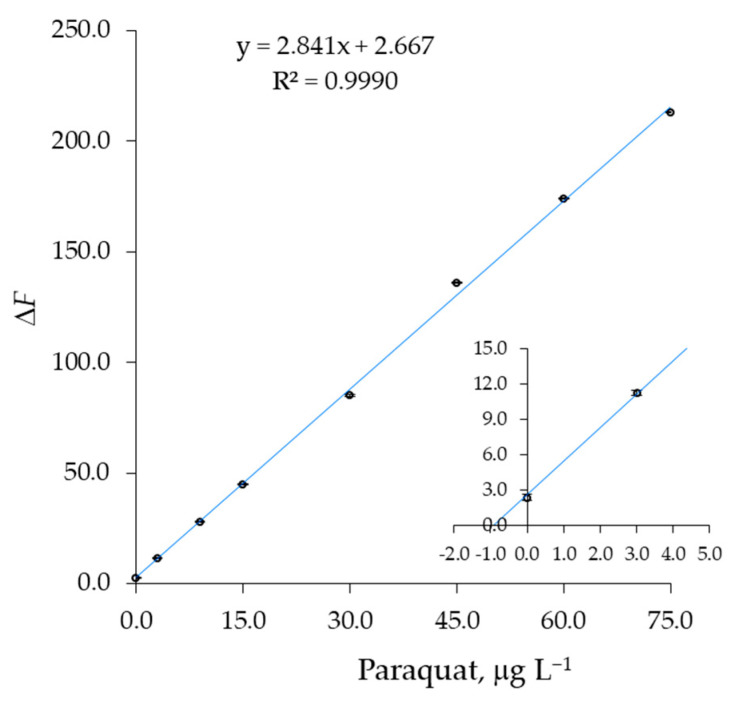
Standard additions plot obtained by adding a methanol solution of paraquat, PQ, and 1b to a tap water sample (3 replicates). The sample was divided into 8 equal aliquots. The spike concentration of 1.0 μg L^−1^ was normalized to the total volume of solvent, 60/40 methanol/water. Inset: The straight line intersects the x-axis to the expected concentration of (1.1 ± 0.1) μg L^−1^ of PQ.

**Figure 12 sensors-23-01120-f012:**
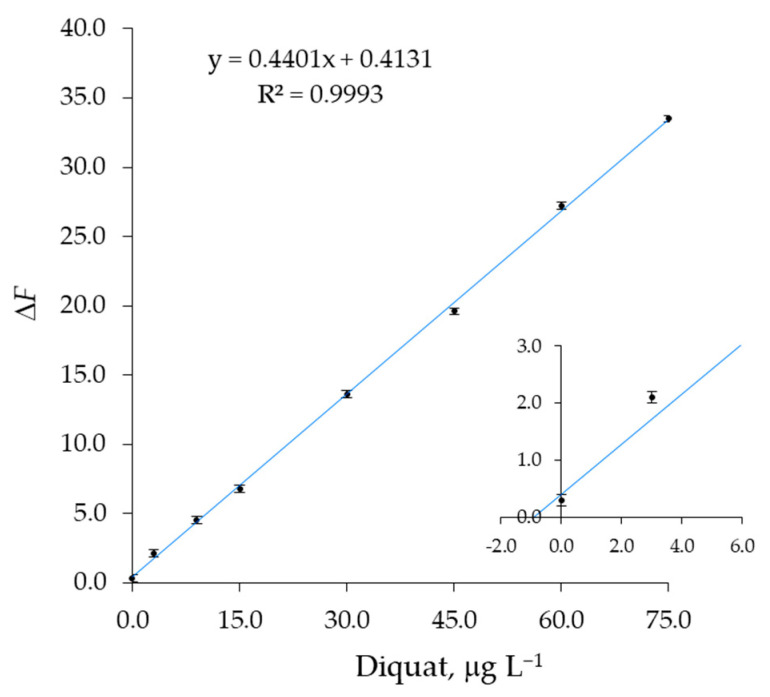
Standard additions plot obtained by adding a methanol solution of diquat, DQ, and 1b to a tap water sample (3 replicates). The sample was divided into 8 equal aliquots. The spike concentration of 1.0 μg L^−1^ was normalized to the total volume of solvent, 60/40 methanol/water. Inset: The straight line intersects the x-axis to the value of (0.9 ± 0.1) μg L^−1^ of diquat.

**Table 1 sensors-23-01120-t001:** Experimental data for the determination of spectra reported in Figure 3. The *V*_1a_ and *V*_PQ_ volumes refer to an initial 1a solution with a concentration of 45.5 μg L^−1^ and paraquat, PQ, solution with a concentration of 1000 μg L^−1^, respectively. The final volume of each solution was made up to a total volume of 3000 μL by fixing with the solvent mixture.

Entry	*V*_1a_, μL	*V*_PQ_, μL	*V*_TOT_, μL	[1a], μg L^−1^	[PQ], μg L^−1^	PQ/1a, mol/mol
1	2000	0	3000	30.3	0.0	0.0
2	2000	2	3000	30.3	0.7	0.1
3	2000	4	3000	30.3	1.3	0.2
4	2000	7	3000	30.3	2.3	0.4
5	2000	11	3000	30.3	3.7	0.6
6	2000	16	3000	30.3	5.3	0.8
7	2000	20	3000	30.3	6.7	1.0
8	2000	27	3000	30.3	9.0	1.4
9	2000	35	3000	30.3	11.7	1.8
10	2000	45	3000	30.3	15.0	2.3
11	2000	53	3000	30.3	17.7	2.7
12	2000	65	3000	30.3	21.7	3.3
13	2000	75	3000	30.3	25.0	3.8
14	2000	92	3000	30.3	30.7	4.7
15	2000	105	3000	30.3	35.0	5.3
16	2000	115	3000	30.3	38.3	5.8
17	2000	132	3000	30.3	44.0	6.7
18	2000	152	3000	30.3	50.7	7.7
19	2000	192	3000	30.3	64.0	9.7
20	2000	252	3000	30.3	84.0	12.7
21	2000	332	3000	30.3	110.7	16.8
22	2000	432	3000	30.3	144.0	21.8
23	2000	572	3000	30.3	190.7	28.9
24	2000	772	3000	30.3	257.3	39.0
25	2000	932	3000	30.3	310.7	47.1

**Table 2 sensors-23-01120-t002:** Experimental data for the determination of spectra reported in Figure 4. The *V*_1b_ and *V*_PQ_ volumes refer to an initial 1b solution with a concentration of 50.7 μg L^−1^ and paraquat, PQ, solution with a concentration of 1000 μg L^−1^, respectively. The final volume of each solution was made up to a total volume of 4000 μL by fixing with the solvent mixture.

Entry	*V*_1b_, μL	*V*_PQ_, μL	*V*_TOT_, μL	[1b], μg L^−1^	[PQ], μg L^−1^	PQ/1b, mol/mol
1	2000	0	4000	25.4	0.0	0.0
2	2000	3	4000	25.4	0.8	0.1
3	2000	6	4000	25.4	1.5	0.3
4	2000	9	4000	25.4	2.3	0.4
5	2000	12	4000	25.4	3.0	0.6
6	2000	16	4000	25.4	4.0	0.8
7	2000	20	4000	25.4	5.0	0.9
8	2000	25	4000	25.4	6.3	1.2
9	2000	32	4000	25.4	8.0	1.5
10	2000	40	4000	25.4	10.0	1.9
11	2000	50	4000	25.4	12.5	2.4
12	2000	60	4000	25.4	15.0	2.8
13	2000	75	4000	25.4	18.8	3.5
14	2000	90	4000	25.4	22.5	4.3
15	2000	110	4000	25.4	27.5	5.2
16	2000	135	4000	25.4	33.8	6.4
17	2000	155	4000	25.4	38.8	7.3
18	2000	185	4000	25.4	46.3	8.7
19	2000	225	4000	25.4	56.3	10.6
20	2000	285	4000	25.4	71.3	13.5
21	2000	365	4000	25.4	91.3	17.3
22	2000	464	4000	25.4	116.0	21.9
23	2000	650	4000	25.4	162.5	30.7
24	2000	800	4000	25.4	200.0	37.8
25	2000	1000	4000	25.4	250.0	47.3

**Table 3 sensors-23-01120-t003:** Experimental data for the determination of spectra reported in Figure 5. The *V*_1a_ and *V*_DQ_ volumes refer to an initial 1a solution with a concentration of 45.5 μg L^−1^ and diquat, DQ, solution with a concentration of 1650 μg L^−1^, respectively. The final volume of each solution was made up to a total volume of 3000 μL by fixing with the solvent mixture.

Entry	*V*_1a_, μL	*V*_DQ_, μL	*V*_TOT_, μL	[1a], μg L^−1^	[DQ], μg L^−1^	DQ/1a, mol/mol
1	2000	0	3000	30.3	0.0	0.0
2	2000	10	3000	30.3	5.5	0.6
3	2000	30	3000	30.3	16.5	1.9
4	2000	50	3000	30.3	27.5	3.1
5	2000	80	3000	30.3	44.0	5.0
6	2000	100	3000	30.3	66.0	7.5
7	2000	120	3000	30.3	66.0	7.5
8	2000	170	3000	30.3	93.5	10.6
9	2000	230	3000	30.3	126.5	14.3
10	2000	300	3000	30.3	165.0	18.7
11	2000	380	3000	30.3	209.0	23.7
12	2000	470	3000	30.3	258.5	29.3
13	2000	570	3000	30.3	313.5	35.5
14	2000	660	3000	30.3	363.0	41.1
15	2000	780	3000	30.3	429.0	48.6
16	2000	870	3000	30.3	478.5	54.2
17	2000	950	3000	30.3	522.5	59.2
18	2000	1000	3000	30.3	550.0	62.3

**Table 4 sensors-23-01120-t004:** Experimental data for the determination of spectra reported in Figure 6. The *V*_1b_ and *V*_DQ_ volumes refer to an initial 1b solution with a concentration of 50.7 μg L^−1^ and diquat, DQ, solution with a concentration of 1650 μg L^−1^, respectively. The final volume of each solution was made up to a total volume of 4000 μL by fixing with the solvent mixture.

Entry	*V*_1b_, μL	*V*_DQ_, μL	*V*_TOT_, μL	[1b], μg L^−1^	[DQ], μg L^−1^	DQ/1b, mol/mol
1	2000	0	4000	25.4	0.0	0.0
2	2000	2	4000	25.4	0.8	0.1
3	2000	5	4000	25.4	2.1	0.3
4	2000	9	4000	25.4	3.7	0.5
5	2000	14	4000	25.4	5.8	0.8
6	2000	20	4000	25.4	8.3	1.2
7	2000	28	4000	25.4	11.6	1.6
8	2000	39	4000	25.4	16.1	2.3
9	2000	54	4000	25.4	22.3	3.1
10	2000	74	4000	25.4	30.5	4.3
11	2000	104	4000	25.4	42.9	6.1
12	2000	144	4000	25.4	59.4	8.4
13	2000	204	4000	25.4	84.2	11.9
14	2000	284	4000	25.4	117.2	16.6
15	2000	384	4000	25.4	158.4	22.4
16	2000	504	4000	25.4	207.9	29.4
17	2000	654	4000	25.4	269.8	38.1
18	2000	854	4000	25.4	352.3	49.8
19	2000	1254	4000	25.4	517.3	73.1
20	2000	1574	4000	25.4	649.3	91.8

**Table 5 sensors-23-01120-t005:** Association constants of H-G complexes at 25 °C, reported as log*K*. Paraquat, PQ, and diquat, DQ, are the guests, G, and the compounds 1a and 1b are the fluorescent hosts, H. The values were determined by Equation (1). Uncertainties are given as standard deviation.

Guest	Host	
	1a	1b
DQ	6.3 ± 0.3	6.7 ± 0.1
PQ	7.1 ± 0.1	7.3 ± 0.1

**Table 6 sensors-23-01120-t006:** Association constants of H-G complexes at 25 °C, reported as log*K*. Paraquat, PQ, and diquat, DQ, are the guests, G, and the compounds 1a and 1b are the fluorescent hosts, H. The values were determined by Equation (3). Uncertainties are given as standard deviation.

Guest	Host	
	1a	1b
DQ	6.4 ± 0.2	6.8 ± 0.1
PQ	7.2 ± 0.1	7.3 ± 0.1

**Table 7 sensors-23-01120-t007:** Detection limits calculated for paraquat, PQ, and diquat, DQ, by fluorescence titration.

Guest	Host	*m*	Detection Limit, ng L^−1^	λ_max_, nm
DQ	1a	1.3577	464	336
DQ	1b	3.8479	164	416
PQ	1a	6.5556	96	336
PQ	1b	20.2826	31	416

**Table 8 sensors-23-01120-t008:** Experimental data for the standard additions plot reported in Figure 11. *V*_1b_ and *V*_PQ_ are the added volumes of the standard methanol solutions of [1b] = 2500 μg L^−1^ and [PQ] = 1500 μg L^−1^, respectively. A tap water sample was spiked and divided into 8 aliquots. The spike concentration of 1.0 μg L^−1^ was normalized to the total volume of 2500 μL, 60/40 methanol/water.

Entry	*V*_sample_, μL	*V*_1b_, μL	*V*_PQ_, μL	*V*_TOT_, μL	[1b], μg L^−1^	[PQ], μg L^−1^	Δ*F* *
1	1000	25	0	2500	25.0	0.0	2.4
2	1000	25	5	2500	25.0	3.0	11.3
3	1000	25	15	2500	25.0	9.0	28.1
4	1000	25	25	2500	25.0	15.0	44.8
5	1000	25	50	2500	25.0	30.0	85.0
6	1000	25	75	2500	25.0	45.0	136.0
7	1000	25	100	2500	25.0	60.0	174.1
8	1000	25	125	2500	25.0	75.0	212.9

* The fluorescence change Δ*F* = *F*_0_ − *F* was calculated with respect to the *F*_0_ value measured in a non-spiked water sample obtained as in entry 1, but from ultrapure, HPLC-grade water. The fluorescence values were measured at 416 nm for anthracene fluorophore 1b. Δ*F* was calculated as an average value of three independent determinations.

**Table 9 sensors-23-01120-t009:** Experimental data for the standard additions plot reported in Figure 12. *V*_1b_ and *V*_DQ_ are the added volumes of the standard methanol solutions of [1b] = 2500 μg L^−1^ and [DQ] = 1500 μg L^−1^, respectively. A tap water sample was spiked and divided into 8 aliquots. The spike concentration of 1.0 μg L^−1^ was normalized to the total volume of 2500 μL, 60/40 methanol/water.

Entry	*V*_sample_, μL	*V*_1b_, μL	*V*_DQ_, μL	*V*_TOT_, μL	[1b], μg L^−1^	[DQ], μg L^−1^	Δ*F* *
1	1000	25	0	2500	25.0	0.0	0.3
2	1000	25	5	2500	25.0	3.0	2.1
3	1000	25	15	2500	25.0	9.0	4.5
4	1000	25	25	2500	25.0	15.0	6.8
5	1000	25	50	2500	25.0	30.0	13.6
6	1000	25	75	2500	25.0	45.0	19.6
7	1000	25	100	2500	25.0	60.0	27.2
8	1000	25	125	2500	25.0	75.0	33.5

* The fluorescence change Δ*F* = *F*_0_ − *F* was calculated with respect to the *F*_0_ value measured in a non-spiked water sample obtained as in entry 1, but from ultrapure, HPLC-grade water. The fluorescence values were measured at 416 nm for anthracene fluorophore 1b. Δ*F* was calculated as the average value of three independent determinations.

**Table 10 sensors-23-01120-t010:** Inorganic species detected in the tested sample of tap water and their effect on the determination of herbicide in a solution of these ions with a 10-fold higher concentration.

Species	Amount, mg L^−1^ of Tested Sample	Addition ^1^ for 10.0 mL	Signal Variation, %
Ca^2+^	42.1	10.52 mg, as CaCO_3_	2
Mg^2+^	12.3	4.89 mg, as MgCl_2_	2
Zn^2+^	0.8	0.35 mg, as ZnSO_4_·7H_2_O	4
Na^+^	39.4	0.70 mg, as NaCl	<1
		3.23 mg, as Na_2_SO_4_	
		0.03 mg, as Na_3_PO_4_	
		6.03 mg, as Na_2_CO_3_	
K^+^	3.1	0.81 mg, as KNO_3_	<1
SO_4_^2−^	23.0	3.23 mg, as Na_2_SO_4_	3
		0.35 mg, as ZnSO_4_·7H_2_O	
PO_4_^3−^	0.2	0.03 mg, as Na_3_PO_4_	2
NO_3_^−^	5.0	0.81 mg, as KNO_3_	4
CO_3_^2−^	98.0	10.52 mg, as CaCO_3_	4
		6.03 mg, as Na_2_CO_3_	
Cl^−^	40.3	4.89 mg, as MgCl_2_	3
		0.70 mg, as NaCl	

^1^ To evaluate the effect of the cations, EDTA was not added, also for solubility problems.

**Table 11 sensors-23-01120-t011:** Experimental data for the selectivity evaluation of the PQ-1b probe in the presence of diquat as interferent. The signal variation was calculated in the linear range from 1.0 to 10.0 μg L^−1^ according to Equation (6), comparing *F*_0_, the maximum fluorescence intensity of a paraquat solution, with *F*, the maximum fluorescence intensity of an equimolar solution of paraquat and diquat. The fluorescence values were measured at 416 nm for anthracene fluorophore 1b.

Entry	[1b], μg L^−1^	[PQ], μg L^−1^	*F* _0_	[DQ], μg L^−1^	*F*	Signal Variation, %
1	25.0	0.0	418.0	0.0	418.0	0.0
2	25.0	1.0	396.5	1.0	392.4	1.0
3	25.0	2.0	376.4	2.0	373.2	0.9
4	25.0	3.0	356.2	3.0	353.5	0.8
5	25.0	5.0	315.5	5.0	312.7	0.9
6	25.0	7.0	275.1	7.0	273.0	0.8
7	25.0	10.0	214.3	10.0	212.8	0.7

**Table 12 sensors-23-01120-t012:** Experimental data for the evaluation of cross-interferences by using the standard addition method. *V*_1b_, *V*_DQ_ and *V*_PQ_ are the added volumes of the standard methanol solutions of [1b] = 2500 μg L^−1^, [DQ] = 3000 μg L^−1^ and [PQ] = 1500 μg L^−1^, respectively. *V*_PQ-spike_ and *V*_DQ-spike_ are the volumes added to a tap water sample to spike it at the concentration of 6.0 μg L^−1^. This value was normalized to the total volume of 2500 μL, 60/40 methanol/water. The sample was divided into 10 aliquots.

Entry	*V*_sample_, μL	*V*_1b_, μL	*V*_PQ-spike_, μL	*V*_DQ-spike_, μL	*V*_DQ_, μL	*V*_TOT_, μL	[1b], μg L^−1^	[PQ]^spike^, μg L^−1^	[DQ]^spike^, μg L^−1^	[DQ], μg L^−1^	Δ*F* *
0	1000	30	0	0	0	2500	30.0	0.0	0.0	0.0	0.0
1	1000	30	10	5	0	2500	30.0	6.0	6.0	0.0	17.9
2	1000	30	10	5	5	2500	30.0	6.0	6.0	6.0	18.2
3	1000	30	10	5	15	2500	30.0	6.0	6.0	18.0	19.2
4	1000	30	10	5	30	2500	30.0	6.0	6.0	36.0	21.9
5	1000	30	10	5	50	2500	30.0	6.0	6.0	60.0	29.1
6	1000	30	10	5	75	2500	30.0	6.0	6.0	90.0	41.9
7	1000	30	10	5	100	2500	30.0	6.0	6.0	120.0	55.6
8	1000	30	10	5	125	2500	30.0	6.0	6.0	150.0	68.4
9	1000	30	10	5	150	2500	30.0	6.0	6.0	180.0	81.7

* The fluorescence change Δ*F* = *F*_0_ − *F* was calculated with respect to the *F*_0_ value measured in a non-spiked water sample obtained mixing the hydroalcoholic solution of 1b with ultrapure, HPLC-grade water (entry 0). The fluorescence values were measured at 416 nm for anthracene fluorophore 1b.

## Data Availability

Not applicable.

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
