# Peer review of "Fluorescence Detecting of Paraquat and Diquat Using Host–Guest Chemistry with a Fluorophore-Pendant Calix[6]arene"

_sensors, 2023, doi:10.3390/s23031120_

Round 1
Reviewer 1 Report
In the paper of Caruso et al. the development of a new fluorescent sensing method for the determination of two herbicides Paraquat and Diquat, is reported. The recognition of these herbicides, which are widely used in agriculture and are contaminants of drinking waters, occurs as a result of host-guest interactions between Calix[6]arene derivatives containing anthracene or naphthalene as a pendant fluorophore and Paraquat e Diquat in hydroalcoholic solution, with a wide linear concentration response (mL-1 ). This is a rigorous study developed with seriousness and accuracy. The experimental procedures and calculation methods used are suitable. Reference have been chosen appropriately. The proposed method has been accurately validated and has been applied for the determination of Paraquat in drinking water samples. The information provided and the conclusion reached are likely and useful for a broad audience, mostly for researchers that work in the environmental field. I suggest that this paper can be published in Sensors as it stands.
Author Response
We appreciate the Reviewer for your precious time in reviewing our paper. We thank the Reviewer for your comments.
Reviewer 2 Report
In this manuscript, the author reported two calix[6]arene derivatives for the determination of paraquat(PQ) and diquat(DQ) in tap and drinking waters. This work seems interesting and has some scientific merits. This manuscript is recommended for possible publication in Sensors after major revision.
1. In Abstract, detection limit and detection ranges can be described specifically.
2. Fluorescence quenching should be used as the keyword instead of fluorescence enhancement.
3. In Introduction, the authors should briefly state the advantages of fluorescence spectroscopy over other methods.
4. The detection principle can be expressed intuitively by drawing a picture.
5. A space should be added between the number and the unit, such as in Figure 3, “25°C” should be “25 °C”.
6. The variables in this manuscript need to be presented in italics. In addition, the figures in this manuscript are irregular, “0,9993” should be “0.9993”.
7. The authors should add some selective experiments to prove that calix[6]arene derivative can only interact with PQ or DQ.
8. There are several grammatical mistakes in this manuscript. For example, in Introduction, “is somewhat” should be “are somewhat”; in Figure 9, “as a guests” should be “as guests”. The author should check it carefully.
Author Response
- In Abstract, detection limit and detection ranges can be described specifically.
Response:
We thank you for this suggestion. The linear response ranges and the detection limits are now reported in the Abstract section.
- Fluorescence quenching should be used as the keyword instead of fluorescence enhancement.
Response:
This change has been made in the Keywords line.
- In Introduction, the authors should briefly state the advantages of fluorescence spectroscopy over other methods.
Response:
We have changed one sentence in the “Introduction section” to address the reviewer’s comment. The text:
Fluorescence spectroscopy is known for its extraordinary sensitivity, high specificity, selectivity, speed, simplicity, and low cost as compared to other analytical techniques [17]. Due to the high levels of sensitivity and in particular their ability to be used for temporal and spatial sampling for in vivo imaging applications, fluorescent chemosensors based on molecular recognition have been widely applied in a variety of fields such as biology, physiology, pharmacology, and environmental sciences [18,19].”
was added in place of the text:
Fluorescence chemosensors based on molecular recognition provide a powerful combination in terms of sensitivity, speed, and portability [17–19].
Consequently, the references [17] and [18] were switched.
- The detection principle can be expressed intuitively by drawing a picture.
Response:
This picture has been included in the submitted graphical abstract.
- A space should be added between the number and the unit, such as in Figure 3, “25°C” should be “25 °C”.
Response:
We deeply apology for this error and corrected it.
- The variables in this manuscript need to be presented in italics. In addition, the figures in this manuscript are irregular, “0,9993” should be “0.9993”.
Response:
We deeply apology for these errors. Thank you. The proposed corrections have been made.
- The authors should add some selective experiments to prove that calix[6]arene derivative can only interact with PQ or DQ.
Response:
The calix[6]arene derivatives interact with the “quat” herbicides as reported in Table 2. The binding constant 1:1 of PQ-host is higher than that of DQ-host probably due to the planar structure of PQ compared to DQ.* Potential cross interferences were evaluated by using the equation 6, comparing F0, the maximum fluorescence intensity of a paraquat solution, with F, the maximum fluorescence intensity of an equimolar solution of paraquat and diquat.
We added the experimental data (Table 8)** to demonstrate that diquat does not interfere with the paraquat determination. In the linear range from 1.0 to 10.0 µg L-1, no significant interferences were observed probably because the interaction for the 1b-PQ probe is preferred (Table 2).
Furthermore, a large excess of DQ added with respect to the PQ shifts the equilibria towards formation of host-guest favored by entropy. The standard addition method leads to good precision, as demonstrated for the most unfavorable case, i.e., the determination of diquat in the presence of paraquat (Table 9). The extrapolated value is in very good agreement with the expected value
* Calculations are being made to determine the stacking interaction between the aromatic units of calix and herbicide guests, as reported in the more appropriate Ref. 38.
** Therefore, previous Table 8 became Table 9.
- There are several grammatical mistakes in this manuscript. For example, in Introduction, “is somewhat” should be “are somewhat”; in Figure 9, “as a guests” should be “as guests”. The author should check it carefully.
Response:
The manuscript has been revised and corrections have been made. We have removed the English mistakes in several part of the paper. We hope it is now correct.
Reviewer 3 Report
In this manuscript, the authors describe the fluorescence detection of herbicides such as Paraquat (PQ) and Diquat (DQ) using a calix[6]arene derivatives (1a and 1b) containing naphthalene or anthracene as a pendant fluorophore. Hydroalcoholic solutions of calix[6]arene derivatives were successfully used to detect these herbicides at very low concentrations by host host-guest molecular recognition method. The fluorescence quenching showed a broad linear response at the μg L-1 of these herbicides’ concentrations with detection limits of the order of ng L-1. Interestingly, the authors successfully used the fluorescence quenching of calix[6]arene derivatives for the determination of paraquat in drinking water samples. Overall, The manuscript is interesting, all the methods are described very well, and the conclusions are supported by experimental results. I recommend this manuscript for consideration for publication in Sensors.
Minor suggestions to authors.
i) As per the discussion, it’s better to remove the word “DQ” in line 111 (or) write it as “(Figures 3 and 4 for PQ and Figures 5 and 6 for DQ)” in lines 112-113.
ii) Show figures 5 and 6, until maximum fluorescence intensity on the Y- axis (or) in other words, cut the Y -axis of Figure 5 at 260 (a.i) and Figure 6 at 340 (a.i).
iii) If possible, in figures 3,4,5, and 6, show which analyte (host) is under study as PQ or DQ on the arrow which represents a decrease of fluorescence intensity of Calix (6) arene.
iv) In figures 11 and 12, place the Y-axis label in the center of the axis.
Author Response
I) As per the discussion, it’s better to remove the word “DQ” in line 111 (or) write it as “(Figures 3 and 4 for PQ and Figures 5 and 6 for DQ)” in lines 112-113.
Response:
We agree with this remark, and we limited the use of PQ and DQ abbreviations accordingly.
II) Show figures 5 and 6, until maximum fluorescence intensity on the Y- axis (or) in other words, cut the Y -axis of Figure 5 at 260 (a.i) and Figure 6 at 340 (a.i).
Response:
The proposed corrections have been made.
III) If possible, in figures 3,4,5, and 6, show which analyte (host) is under study as PQ or DQ on the arrow which represents a decrease of fluorescence intensity of Calix (6) arene.
Response:
These labels have been added in these figures.
IV) In figures 11 and 12, place the Y-axis label in the center of the axis.
Response:
These specific corrections have been made to the manuscript, even in accord to the Reviewer 2 requests.
Round 2
Reviewer 2 Report
The authors have made corresponding modifications to the original article according to the comments. There are still minor problems (e.g., Figure 10, “R2” should have the same number of decimal place), which can be revised in the process of proof-checking.